# Kinetics of Renal Function during Induction in Newly Diagnosed Multiple Myeloma: Results of Two Prospective Studies by the German Myeloma Study Group DSMM

**DOI:** 10.3390/cancers13061322

**Published:** 2021-03-16

**Authors:** Friederike Bachmann, Martin Schreder, Monika Engelhardt, Christian Langer, Denise Wolleschak, Lars Olof Mügge, Heinz Dürk, Kerstin Schäfer-Eckart, Igor Wolfgang Blau, Martin Gramatzki, Peter Liebisch, Matthias Grube, Ivana v Metzler, Florian Bassermann, Bernd Metzner, Christoph Röllig, Bernd Hertenstein, Cyrus Khandanpour, Tobias Dechow, Holger Hebart, Wolfram Jung, Sebastian Theurich, Georg Maschmeyer, Hans Salwender, Georg Hess, Max Bittrich, Leo Rasche, Annamaria Brioli, Kai-Uwe Eckardt, Christian Straka, Swantje Held, Hermann Einsele, Stefan Knop

**Affiliations:** 1Division of Nephrology and Medical Intensive Care, Charité University Medicine, 10117 Berlin, Germany; friederike.bachmann@charite.de (F.B.); kai-uwe.eckardt@charite.de (K.-U.E.); 2Division of Hematology and Oncology, Würzburg University Hospital Medical Center, 97080 Würzburg, Germany; martin.schreder@gesundheitsverbund.at (M.S.); bittrich_m@ukw.de (M.B.); rasche_l@ukw.de (L.R.); Einsele_H@ukw.de (H.E.); 3Freiburg University Hospital, 79106 Freiburg, Germany; monika.engelhardt@uniklinik-freiburg.de; 4Ulm University Hospital, 89070 Ulm, Germany; christian.langer@klinikverbund-allgeau.de; 5Magdeburg University Hospital, 39120 Magdeburg, Germany; denise.wolleschak@med.ovgu.de; 6Jena University Hospital, 07747 Jena, Germany; lars-olof.muegge@hbk-zwickau.de (L.O.M.); Annamaria.Brioli@med.uni-jena.de (A.B.); 7St. Barbara- Hospital Hamm GmbH, 59075 Hamm, Germany; heinz.duerk@josef-krankenhaus.de; 8Nuremberg Hospital, Paracelsus Medizinische Privatuniversität, 90419 Nuremberg, Germany; kerstin.schaefer-eckart@klinikum-nuernberg.de; 9Division of Hematology and Immunology, Charité University Medicine, 13353 Berlin, Germany; igor.blau@charite.de; 10Schleswig-Holstein University Hospital, Kiel Campus, 24105 Kiel, Germany; martin.gramatzki@uksh.de; 11Private Oncology Practice, 47441 Moers, Germany; liebisch@onkologie-moers.de; 12Regensburg University Hospital, 93043 Regensburg, Germany; matthias.grube@klinik.uni-regensburg.de; 13Frankfurt University Hospital, 60590 Frankfurt, Germany; ivana.metzler@kgu.de; 14University Hospital re. d. Isar, 81675 Munich, Germany; florian.bassermann@mri.tum.de; 15Oldenburg University Hospital, 26133 Oldenburg, Germany; bernd.metzner@t-online.de; 16Dresden Carl Gustav Carus University Hospital, 01307 Dresden, Germany; christoph.roellig@uniklinikum-dresden.de; 17Bremen Municipal Hospital Mitte, 28205 Bremen, Germany; Bernd.Hertenstein@klinikum-bremen-mitte.de; 18Munster University Hospital, 48149 Munster, Germany; cyrus.khandanpour@ukmuenster.de; 19Private Oncology Practice, 88212 Ravensburg, Germany; tobias.dechow@onkonet.eu; 20Stauferklinikum, Schwäbisch Gmünd, 73557 Mutlangen, Germany; holger.hebart@kliniken-ostalb.de; 21Göttingen University Hospital, 37099 Göttingen, Germany; wolfram.jung@med.uni-goettingen.de; 22Munich Großhadern University Hospital, 80336 Munich, Germany; sebastian.theurich@med.uni-muenchen.de; 23Ernst von Bergmann Hospital, 14467 Potsdam, Germany; georg.maschmeyer@evbk.de; 24Asklepios Klinikum Altona, 22763 Hamburg, Germany; h.salwender@asklepios.com; 25Mainz University Hospital, 55131 Mainz, Germany; georg.hess@unimedizin-mainz.de; 26Munich Hospital Schwabing, 80804 Munich, Germany; christian.straka@muenchen-klinik.de; 27Department of Biostatistics at ClinAssess GmbH, 51379 Leverkusen, Germany; swantje.held@clinassess.de

**Keywords:** multiple myeloma, renal failure, kidney, bortezomib, lenalidomide, induction regimen

## Abstract

**Simple Summary:**

Renal insufficiency is frequently seen in newly diagnosed multiple myeloma and can be due to the disease itself but also caused by medical interventions or infections. Patients with severe renal insufficiency are known to have an adverse prognosis, but recently, it was shown that even moderately impaired kidney function can have long-term sequelae. Achieving quick disease control by effective antimyeloma therapy can lead to the recovery of renal function. We investigated the kidney-specific variables in a large cohort of 770 myeloma patients receiving three different three-drug regimens for initial myeloma treatment to learn more about the differential effects on kidney function in an early disease phase. All regimens had a positive impact on kidney function without a difference in the proportion of patients who reached normal renal function after three cycles. Interestingly, patients who received bortezomib, lenalidomide, and dexamethasone tended to have higher risk for a worse renal function following induction when compared to the initial values.

**Abstract:**

Background: Preservation of kidney function in newly diagnosed (ND) multiple myeloma (MM) helps to prevent excess toxicity. Patients (pts) from two prospective trials were analyzed, provided postinduction (PInd) restaging was performed. Pts received three cycles with bortezomib (btz), cyclophosphamide, and dexamethasone (dex; VCD) or btz, lenalidomide (len), and dex (VRd) or len, adriamycin, and dex (RAD). The minimum required estimated glomerular filtration rate (eGFR) was >30 mL/min. We analyzed the percent change of the renal function using the International Myeloma Working Group (IMWG) criteria and Kidney Disease: Improving Global Outcomes (KDIGO)-defined categories. Results: Seven hundred and seventy-two patients were eligible. Three hundred and fifty-six received VCD, 214 VRd, and 202 RAD. VCD patients had the best baseline eGFR. The proportion of pts with eGFR <45 mL/min decreased from 7.3% at baseline to 1.9% PInd (*p* < 0.0001). Thirty-seven point one percent of VCD versus 49% of VRd patients had a decrease of GFR (*p* = 0.0872). IMWG-defined “renal complete response (CRrenal)” was achieved in 17/25 (68%) pts after VCD, 12/19 (63%) after RAD, and 14/27 (52%) after VRd (*p* = 0.4747). Conclusions: Analyzing a large and representative newly diagnosed myeloma (NDMM) group, we found no difference in CRrenal that occurred independently from the myeloma response across the three regimens. A trend towards deterioration of the renal function with VRd versus VCD may be explained by a better pretreatment “renal fitness” in the latter group.

## 1. Introduction

The kidney is an important target organ in plasma cell dyscrasias. Besides a wide spectrum of clinical presentations, there are also multiple underlying etiologies that may affect different anatomical kidney structures [1]. Physicochemical features of a secreted monoclonal protein probably are the most important determinants as to whether renal disease will ultimately develop [2]. In multiple myeloma (MM), acute kidney injury (AKI) carries the worst prognosis [3], with some extent of renal involvement being present in at least 30% of patients [4]. The prototypical lesion associated with AKI is light chain cast nephropathy [5,6], which may be fully reversible, a phenomenon called “renal recovery” [7,8]. Renal recovery is a dramatically improved organ function is due to effective suppression of the underlying clone [9,10,11,12]. Precipitating (or independent) factors may be arterial hypertension and diabetes mellitus but, also, advanced age, anemia, myeloma-related factors (infection and hypercalcemia), and drugs (NSAIDs and contrast media) [13,14]. In contrast to a phase 3 study for nontransplant-eligible subjects [15], most recent trials involving high-dose chemotherapy (HDT) and autologous stem cell transplant (ASCT) applied strict criteria for renal function. In young and medically fit patients, the latter approach remains the standard of care despite the introduction of novel compounds [16,17]. In a trial of various post-HDT strategies, as well as in one investigating the addition of the anti-CD38 antibody, daratumumab, to bortezomib, thalidomide, and dexamethasone, a creatinine clearance of at least 40 mL/min was required [18,19]. In a comparison of bortezomib, lenalidomide, and dexamethasone (VRD) with versus without HDT/ASCT, participants needed a minimum glomerular filtration rate (GFR) as high as 50 mL/min [16] to enroll. The International Myeloma Working Group (IMWG) has issued two consensus papers detailing the definitions and pathophysiology of renal involvement and has highlighted the “renal recovery” concept based on suppression of the clone(s) [20,21]. Specifically, the criteria of a “renal response to antimyeloma therapy” have been widely adopted to measure the effects in MM patients with preexisting renal failure. Very recently, a negative impact of long-term outcomes in transplant-eligible patients was, for the first time, linked to only moderately decreased estimated glomerular filtration rate (eGFR) [22]. Relatively little is known on the early effects of modern induction triplets in patients with mild-to-moderately impaired renal function, i.e., in the range between 60- and 30-mL/min eGFR. While bortezomib has proven effective in the reversal of renal failure [9,12,23], some concerns exist about the detrimental effects of lenalidomide, such as Fanconi syndrome, azotemia, and interstitial nephritis [24,25,26]. We set out to compare pre- and postinduction renal function, adverse effects and the antimyeloma response across three different induction triplets from two prospective studies by the German Multiple Myeloma Study Group (DSMM) XI and XIV. As the evidence suggests that the GFR category “>30 and <60 mL/min” is exceptionally heterogeneous in terms of outcomes and risk profiles in noncancer patients [27], we decided to also use the KDIGO criteria [21].

## 2. Patients and Methods

### 2.1. Study Design and Patients

Briefly, the DSMM XI study (EudraCT 2005-003902-27) was a phase 2 study evaluating bortezomib, cyclophosphamide, and dexamethasone (VCD) prior to planned tandem HDT/ASCT in patients with newly diagnosed myeloma (NDMM) up to the age of 60 years. Cyclophosphamide dose was 900 mg/m^2^ intravenously when administered on cycle day 1 every three weeks. Bortezomib 1.3 mg/m^2^ was administered intravenously on cycle days 1, 4, 8, and 11 and 40-mg oral dexamethasone on cycle days 1/2, 4/5, 8/9, 11, and 12) [28]. Restaging, and end of study, was scheduled for day 63, calculated from the initial doses of cyclophosphamide and bortezomib/dex. Progression-free or overall survival data are not available from this induction-only protocol.

The DSMM XIV study was a phase 3 trial (EudraCT 2009-016616-21) that included NDMM patients up to 65 years of age. It was designed corresponding to a double 2-by-2 factorial design. The first randomization tested the noninferiority of a lenalidomide-based induction triplet (three cycles of lenalidomide, adriamycin, and dexamethasone (RAD) [29] when compared to three cycles of bortezomib, lenalidomide, and dexamethasone (VRd). RAD consisted of lenalidomide 25 mg, days 1–21, adriamycin intravenous push on days 1–4, and oral dexamethasone 40 mg, days 1–4 and 17–20 of a four-week cycle. The VRd regimen comprised a subcutaneous bortezomib on days 1, 4, 8, and 11; lenalidomide 25 mg days 1–14; and dexamethasone 20 mg on cycle days 1/2, 4/5, 8/9, 11, and 12 of a three-week cycle. Patients underwent a first complete restaging procedure following three cycles of RAD or VRd, respectively.

### 2.2. Analysis of Renal Parameters and Adverse Effects

Assuming that the cumulative dose of myeloma-directed treatment is linked to an improvement of renal impairment, we only analyzed patients who received all three preplanned induction cycles. The primary efficacy endpoint of this post-hoc analysis of the two clinical studies was the improvement of previously impaired renal function, most often referred to as “renal recovery”.

Renal recovery is defined as an increase of the estimated GFR (eGFR) during induction therapy, which is reflected by the difference between the eGFR of patients with moderately impaired renal function (eGFR < 45 mL/min/1.73 m^2^) at screening and eGFR after induction therapy. The protocol-defined inclusion criterion regarding baseline renal function in the DSMM XI and DSMM XIV protocols was an eGFR of at least 30 mL/min/1.73 m^2^. As the criteria suggested by Dimopoulos and coworkers are widely accepted, their definition of “renal complete response (CRrenal)” was used as well: average baseline eGFR <50 mL/min/1.73 m^2^ reaching values ≥60 mL/min/1.73 m^2^. For a more differentiated approach, we decided to use the categories according to the KDIGO 2012 nomenclature [27]:eGFR > 60 mL/min/1.73 m^2^ (here, further referred to as “group I”, equivalent to KDIGO G1 and 2).eGFR > 45 and <60 mL/min/1.73 m^2^ (“group II”, equivalent to KDIGO G3a).eGFR >30 and <45 mL/min/1.73 m^2^ (“group III”, equivalent to KDIGO G3b).

Additionally, as a noncategorical approach, we used the relative percent change of eGFR postinduction when compared to the baseline. The statistical analysis was performed using SAS^®^ version 9.4 (SAS Institute Inc., Cary, NC, USA). The severity of adverse events (AEs) was graded by the investigators according to NCI-CTCAE version 3.0 (DSMM XI) and version 4.0 (DSMM XIV), respectively. Recorded AEs were be coded by preferred term and system organ class using MedDRA version 18.1 (for both DSMM XI and DSMM XIV). The study objectives were as follows: (i) to compare renal response/recovery to three different induction regimens (VCD versus RAD versus VRD) in NDMM patients with and without relevant nephrological comorbidities such as diabetes mellitus and hypertension, (ii.) to correlate renal recovery with the depth of remission of MM and with overall survival, and (iii.) to evaluate nephrotoxicity of the respective induction protocols measured by a decrease of eGFR from screening to restaging and by the incidence and severity of AEs associated with kidney function.

### 2.3. Statistical Considerations

Standard descriptive methods were used to present all relevant data. Continuous data were summarized with the following items: number of non-missing observations, missing observations, arithmetic mean, standard deviation, median, and extremes. Categorical data were presented in contingency tables with frequencies and percentages of each modality (including missing data modality).

The assumed normal distribution of the parameter “eGFR“ was tested according to Kolmogorov-Smirnov and the Shapiro-Wilk tests and, in addition, was graphically reviewed via Q-Q (quantile-quantile) plots and histograms. ANOVA (analysis of variance) was applied to study the efficacy of the different protocols regarding renal recovery. Therefore, depending on the induction regimens, the improvement of renal function as expressed by “eGFR postinduction therapy—eGFR screening” and its relative change (in percent) was tested for significance by an averaging process. If “eGFR” would prove to be nonparametric, the Kruskal-Wallis test was applied to examine the efficacy of the different treatment arms. The influence of relevant nephrological comorbidities (diabetes mellitus and arterial hypertension) on the recovery of renal impairment was studied by ANOVA or the Kruskal-Wallis test, respectively, choosing the following factors: (i) diabetes (DM), (ii) arterial hypertension (HT), and (iii) induction regimen. We defined a renal response for a given patient as the improvement of eGFR from screening to the value after induction therapy of at least one group and correlated this with the “myeloma response to induction therapy”. For this correlation, the Kendall rank correlation coefficient and Cramér’s V were used, as both variables are scaled categorically. For all other parameters, appropriate nonparametric tests were applied. Two-sided tests with α = 5% were used unless otherwise indicated.

## 3. Results

### 3.1. Patient Characteristics

Overall, 772 subjects were included in the analysis; 356/391 patients (91%) with a median age of 54 (range, 32–62) years from the DSMM XI study received all three induction cycles and were evaluable for a postinduction response. A total of 476 patients were enrolled into the DSMM XIV protocol, 469 were dosed and, thus, constituted the safety set; 232 patients were randomized to receive RAD and 237 VRd, respectively. Two hundred and two RAD patients (87%) with a median age of 55 (range, 32–65) years and 214 (90.3%) VRd patients with a median age of 56 (range, 32–65) years completed the induction as planned. VCD (58.1%), 61% of VRd, and 62.4% of RAD patients were of male gender. The International Staging System (ISS) stage III disease was present in 13.8% of VCD, 16.3% of RAD, and 16.4% of VRd patients (Table 1).

### 3.2. Baseline Renal Function

The median eGFR rate was 95.6 mL/min for VCD, 79.7 mL/min for RAD, and 73.1 mL/min for VRd patients (*p* < 0.0001). Only five patients had baseline eGFRs < 30 mL/min. They were included for further analysis in the lowest eGFR group (>30 and <45 mL/min). Utilizing MDRD IV or CKD-Epi formulas, 56 patients (7.3%) were found in KDIGO group III (lowest estimated GFR: >30 and <45 mL/min). The distribution was as follows: 18 (5.1%) of VCD, 17 (8.4%) of RAD, and 21 (9.8%) of VRd patients were found in this group. In contrast, 78.5% of VRd, 84.7% of RAD, and 87.1% of VCD-treated patients were classified in group I with an eGFR >60 mL/min (*p* = 0.0428). Utilizing the threshold of <50 mL/min (according to the IMWG consensus statement), 74 patients (9.6%) were in this group at the baseline: 25 receiving VCD (7%), 22 receiving RAD (10.9%), and 27 (12.6%) undergoing induction with VRd (Table 2).

### 3.3. Relevant Comorbidities

The observation of a slightly higher a rate of baseline renal impairment in patients scheduled to receive lenalidomide was neither reflected by comorbidities potentially influencing the kidney function nor by the amounts of proteinuria: the proportions of patients with arterial hypertension were 31.5% in the VCD, 35.2% in the RAD, and 29.9% in the VRd groups. Three point four percent of VCD, 5.0% of RAD, and 2.3% of VRd patients had a medical history of both hypertension and diabetes mellitus (Table 1). The rate of patients with more than two antihypertensive drugs was between 11.8% (VCD) and 15.4% (RAD). Only about 1% had insulin-dependent diabetes mellitus. The body mass index values were evenly distributed across the treatment groups: median values were 25.5 for VCD, 26.2 for RAD, and 26.0 for VRd patients. One hundred and eighty-seven patients in the VCD group (52.5%), 157 in the RAD (77.7%), and 167 in the VRd group (78%) were evaluable for proteinuria by urine protein electrophoresis and-immunofixation; 52.6% of VCD, 62.4% of RAD, and 60.3% of VRd patients had some extent of light-chain excretion. The proportion of patients with light-chain proteinuria > 500 mg/24h was 23.9% in the VCD, 5.4% with RAD, and 6.1% with VRd. None of the patients had AL- or light-chain deposition disease-typical albuminuria.

### 3.4. Changes of Renal Function

#### Renal Recovery (IMWG Criteria) and Differential Changes in GFR Values and Categories during Induction

A “CRrenal”, as defined per the 2010 IMWG consensus criteria, occurred in 43/71 of the cases (60.6%). Comparing the three different protocols, CR renal occurred in 17/25 (68%) pts following VCD, 12/19 (63%) following RAD, and 14/27 (52%) following VRd induction (*p* = 0.4747). When further using KDIGO-defined categories, the proportion of patients in the lowest (GFR > 30 and <45 mL/min) category decreased by more than two-thirds, from 7.3% to 1.9% (*p* < 0.0001). Only 1.1% of VCD, 2.0% of RAD, and 3.3% of VRd-treated patients were classified accordingly (Table 3, Table 4 and Table 5). To evaluate the differential longitudinal development of eGFR, we analyzed the relative percent change and categorical increase and decrease to the induction protocols. VCD patients (128/345) (37.1%) experienced a deterioration of their renal function, with a median eGFR loss of −7.7% (range, −9.3–0.0%), while 217 (62.9%) had a stable or increased value: median eGFR gain 20.5% (range, 0.0–350.4%). The RAD subjects (84/194) (43.3%) had a median eGFR decrease of −3.9% (range, −53.9–0.0%), while 110 (56.7%) had a median improvement of 22.5% (range, 1.7–171.9%). With VRd, 96/196 (49%) patients had a median decrease of -3.6% (range, −49.3–0.0%), while 100/196 (51%) improved their eGFR by a median of 21.4% (range, 0.7–186%). When comparing the bortezomib groups, there was a trend towards a higher proportion of patients with a GFR decrease with VRd (*p* = 0.0872), while the percent loss was comparable (*p* = 0.1462). In a category-centric analysis, the proportion of subjects with an increase of one or two categories was the highest with VRd (13.5%) and lowest with VCD (8.8%; *p* = 0.0681). The picture was vice versa when analyzing patients with a deterioration of renal function during induction; while only one VCD-treated subject (0.3%) experienced a downgrade of one KDIGO-defined GFR group, eight (3.7%) patients receiving VRd (*p* = 0.0013 for VCD versus VRd) and five (2.5%) patients following RAD experienced a loss of one KDIGO category (comparison between groups for all steps of improvement/deterioration: *p* = 0.0167). All except one of the 14 patients with a deterioration of their renal function received zoledronic acid in an adjusted dose. No other potentially nephrotoxic drugs, infections of > grade 2, or the administration of contrast media were recorded. The association of pre- and postinduction eGFR values is given in Figure 1.

### 3.5. Correlating Kinetics of Renal Function with Antimyeloma Response and Overall Survival (OS)

Next, we analyzed the relationship between the antimyeloma response and improvement or deterioration, respectively, of the renal function. The overall response rate (i.e., at least a partial response, PR) was 89.3% with VCD, 90.6% with RAD, and 91.1% following VRd, respectively (*p* = 0.7165). An upgrade of at least one in the KDIGO-defined categories occurred in 31/38 (81.6%) of VCD, 24/31 (77.4%) of RAD, and 29/33 (87.9%) of VRD patients, respectively (*p* = 0.1108). We next correlated the probability of any upgrade of the renal function as defined per the IMWG (20) with a postinduction response. Patients (15/17) (88.2%) having at least very good partial response (VGPR) post-VCD induction versus two out of two with PR or less (100%) achieved renal recovery (*p* > 0.9999). In the RAD cohort, 5/12 patients (41.7%) with at least VGPR versus all seven with only PR or less had their renal function improved (*p* = 0.0174). Patients (8/14) (46.2%) with at least VGPR post-VRD induction versus 6/13 (42.9%) with PR or less achieved renal improvement (*p* = 0.7064). For the RAD and VRd cohorts, the overall survival data at the median follow-up of 47.9 months was available. The median OS was not reached with either induction. In RAD-treated patients, the hazard ratio (HR) for the seven patients with versus the 12 without IMWG-defined renal recovery was 2.705 (95% confidence interval (CI), 0.299–24.509; *p* = 0.3572). Following VRD, the HR for the 13 patients with versus the 14 without renal recovery was 0.963 (95% CI, 0.134–6.927; *p* = 0.9699).

### 3.6. Adverse Events

VCD-treated (95/356) (26.7%), 47/202 (23.3%) of RAD-treated, and 67/214 (31.3%) of VRd-treated subjects experienced at least one kidney-associated treatment-emergent adverse events (TE-AE) (*p* = 0.1779). The rates of severe (>grade 3) TE-AEs were 2.8% for VCD, 5.4% for RAD, and 1.9% for VRd induction (*p* = 0.0988). Eleven cases of acute renal failure/decreased GFR (5.4%) were observed with RAD and six (2.8%) with VRd, while only one such event (0.3%) was recorded for VCD (*p* = 0.0005). Six of the eleven cases with RAD and two of the six with VRd were classified as > grade 3. All affected subjects initially had an eGFR of >50 mL/min and ≤70 mL/min, except three with RAD induction who had stable disease (SD), achieved at least a PR. None of the renal injury episodes were associated with hypercalcemia nor a relevant infection. A serum creatinine/blood urea nitrogen increase was documented in seven (2.0%) of VCD, three (1.5%) RAD, and five (2.3%) VRd patients (*p* = 0.8199). Hypertension occurred in 5 (2.5%) of RAD, 6 (2.8%) of VRd, and 20 (5.6%) of VCD patients (*p* = 0.1091).

## 4. Discussion

To gain more insight into the potentially diverging effects of various MM induction regimens, we analyzed the kinetics of the renal function, along with the baseline comorbidities and renal AEs, in a large cohort from two prospective studies involving three different triplets. Triplets for induction are nowadays the mainstay prior to HDT/ASCT, with the VRd regimen being considered as the “new standard” due to its outstanding efficacy [16,18]. As effective antimyeloma treatments can induce the improvement of even severe renal failure [12] and may thus lead to an improved prognosis of affected subjects [30,31], we took advantage of the narrowly overlapping in- and exclusion criteria in two of our studies. In contrast to other reports [16,18,19], the population we analyzed needed an eGFR as low as 30 mL/min to enroll. Nevertheless, patients with KDIGO stages 4 and 5 were not allowed to enroll; thus, the differential impact of the induction analyzed here may not apply to severely compromised subjects. Using the IMWG criteria for the definition of the renal response to antimyeloma therapy [20], 9.2% of patients were found in the baseline GFR category of <50 mL/min. The rates of renal recovery ranged from 68% with VCD and 63% with RAD to only 52% with VRd induction, respectively. Interestingly, in our series, renal recovery was not correlated with the depth of the postinduction myeloma response of at least a VGPR. The probability of renal recovery in the very few patients with PR or less were substantial, even with all seven patients in the RAD cohort achieving this renal endpoint. In the VCD cohort, unfortunately, the criterion of a “sustained response” and time-dependent variables were not available, since the patients were not followed-up on after completed restaging. When applying the KDIGO 2012 criteria, the results varied over a wider range. The rationale for these thresholds to use was that KDIGO categories 3a and 3b (as validated in noncancer patients) have a considerably different prognostic impact. To the best of our knowledge, only one retrospective study in the myeloma field has previously applied these categories so far [32]. The KDIGO thresholds, when used, still put 7.3% of the patients from our cohort into the lowest eGFR group at the baseline, and only 1.9% of the patients were found in this group following induction. While this suggests an overall efficacy of all three regimens, heterogeneity of the renal baseline features needs to be taken into account. Due to its safety profile and metabolism, integrating bortezomib into treatments seems particularly reasonable and, thus, a comparison between VCD and VRd in our study of particular interest. Kumar and colleagues performed a chart review in approximately 175 patients receiving these two regimens and demonstrated some interesting aspects, albeit with a short follow-up. Of note, they did not find any meaningful differences for the antimyeloma efficacy variables, including survival [33]. In a previous abstract, they described around 15% renal-adverse events for either group [34], while this observation was not reported in the full-length publication. When we focused on these two most widely used protocols, we found that the relative gain in GFR was identical with 21%.The maximum GFR improvement (app. 3.5-fold) occurred in the VCD cohort, while the highest increase was 1.8-fold with VRd. The proportion of patients with a decrease in GFR tended to be higher with VRd when compared to VCD, while the extent of the deterioration was comparable. Significantly more VCD- versus VRd-treated patients (96% versus 82% when applying the KDIGO criteria) reached an eGFR equal to or above 60 mL/min. These observations, however, need to be interpreted with caution. The subjects in the VCD cohort had potentially higher “renal fitness” and a better risk profile of their myelomas. However, the prevalence of an at least 500 mg/d of light-chain proteinuria was higher with VCD. As between 15% and 20% of cases of renal recovery occur without an objective myeloma response, one needs to acknowledge that the overall effects of myeloma care were likely to contribute to an improved general condition of the affected subjects. Finally, when looking at treatment-emergent AEs, patients receiving lenalidomide as part of their induction had significantly higher rates of adverse effects than patients with the bortezomib-only treatment. The incidence across all regimens, however, was moderate at 2.3%. Accordingly, most renal toxicities described in conjunction with VRd were of grades 1 and 2 severity [16,33,34]. Our study had some limitations: the overall number of subjects with an eGFR below 45 mL/min was small and thereby limited the generalizability for this particular group. The long-term impact of early renal recovery with either induction protocol could not be assessed, as the phase II VCD study had too short of a follow-up. Furthermore, due to double the rate of VRd patients being in the worst baseline eGFR group (when compared to VCD), a direct comparison of the added efficacy and toxicity derived from lenalidomide was not possible. This also applied to dexamethasone. Given its potency as an antimyeloma drug, the different dosing (320 mg/cycle with RAD and VCD versus 160 with VRd) might have an impact on the overall “renal efficacy” of an induction regimen. A more detailed analysis was prevented by the fact that approximately 60% of the subjects had no full urine protein analyses available. Even though most patients from our study had only mild-to-moderately impaired renal function, a surprising finding by a Polish group underscored the relevance of our results; they demonstrated a shorter progression free survival (PFS) and, also, OS in HDT/ASCT patients entering therapy with an eGFR threshold below 55 mL/min. [22]. While we were not able to reproduce this finding for the RAD and VRd cohorts, their observations joined the results pointing to the early benefit of bortezomib-based induction in two patient cohorts with mild-to-moderately impaired renal function (median eGFR around 50 mL/min [10]). The renal recovery in this series occurred as early as three months from the start of treatment and was substantial still for patients with eGFR < 90 mL/min. Overall, comparable analyses are scarce, making our current study the largest investigation of the early changes of the renal function in transplant-eligible myeloma.

## 5. Conclusions

We analyzed a large and, in terms of the minimum required GFR of 30 mL/min, representative group of NDMM subjects from prospective, multicenter phase 2/3 studies receiving three different induction triplets with respect to the patterns of renal response and adverse events. Widely overlapping in- and exclusion criteria allowed for a successful side-by-side comparison approaching the validity of a “true” randomized study. Applying the KDIGO criteria, the highest rate of patients with a postinduction eGFR of at least 60 mL/min was achieved following VCD. Regarding up- and downgrading of the renal function categories, the risk of deterioration seemed higher with VRd, albeit VCD patients might have come up with a better pretreatment “renal fitness”. Overall, our study did not find a superiority of VRd over VCD and RAD when analyzing a large array of renal parameters. In our series, VCD seemed to cause less decrease in renal function. However, these patients displayed less pretreatment risk for treatment-induced renal impairment. Future, prospective studies are warranted to dissect patients at a higher risk for treatment-associated renal damage from those who will tolerate VRd from the very beginning.

## Figures and Tables

**Figure 1 cancers-13-01322-f001:**
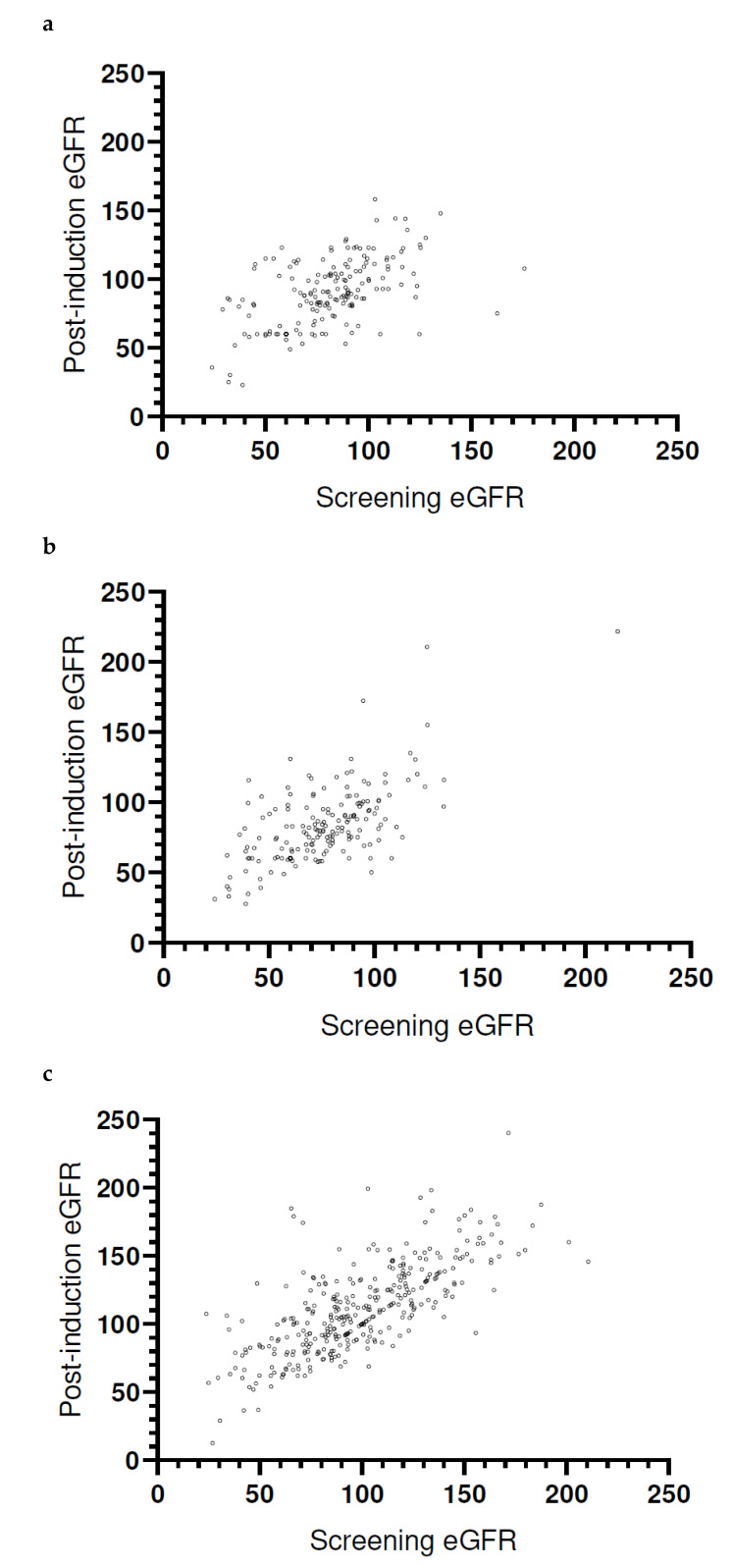
Changes in screening and postinduction GFR categories per induction regimen. (**a**) Two hundred and two patients receiving RAD for induction; postinduction data (not applicable) n.a. for 8 patients (pts). (**b**) Two hundred and fourteen patients receiving VRd for induction; baseline data n.a. in 3, and postinduction n.a. in 17 pts. (**c**) Three hundred and fifty-six patients receiving VCD for induction; baseline data n.a. in 8, and postinduction n.a. in 11 pts. Abbreviations: eGFR, estimated glomerular filtration rate; RAD, lenalidomide, adriamycin, and dexamethasone. VRd, bortezomib, lenalidomide, and dexamethasone. VCD, bortezomib, cyclophosphamide, and dexamethasone.

**Table 1 cancers-13-01322-t001:** Baseline characteristics.

	RAD (*N* = 202)	VRd (*N* = 214)	VCD (*N* = 356)
Median age, y (range)	55 (32–65)	56 (32–65)	54 (32–62)
Male, *N* (%)	143 (62.4)	141 (61.0)	207 (58.1)
ISS stage III, *N* (%)	43 (18.8)	39 (16.9)	49 (13.8)
Type of myeloma, N(%)			
IgA	49 (21.4)	47 (20.4)	71 (19.9)
IgD	3 (1.3)	1 (0.4)	0
IgG	126 (54.0)	131 (54.7)	189 (53.0)
Light chain	20 (8.7)	24 (10.4)	37 (10.4)
Other	17 (7.4)	14 (6.1)	21 (5.9)
Unknown	14 (6.1)	12 (5.2)	38 (10.6)
Median estimated GFR (mL/min(/1.73 m^2^)), range	79.7 (24.1–226.5)	73.1 (23.8–210.7)	95.6 (23.8–210.7)
High-risk abnormalities in FISH diagnostics at baseline, *N* (%)			
del TP53	25 (12.4)	23 (10.7)	27 (7.6)
t(4;14)	24 (11.9)	26 (12.1)	35 (9.8)
t(14;16)	8 (4.0)	9 (4.2)	n.k.
Positive urine immunofixation, *N* (%)	126 (62.4)	129 (60.3)	187 (52.6)
Mean amount of positive patients (mg/24 h)	3,180	2,848	2,100
>500mg/24 h, N(%)	11(5.4)	13 (6.1)	85(23.9)
>200mg albumin/24 h, N(%)	0(0)	0(0)	0(0)
not done, N(%)	5(2.4)	47(19.5)	169(47.4)
Median BMI, kg/m^2^ (range)	26.2 (17.6–43.6)	26.0 (16.2–45.9)	25.5 (16.4–44.9)
Hypertension, *N* (%)	72 (35.6)	64 (29.9)	112 (31.5)
> 2 antihypert. drugs	18 (8.9)	20 (9.3)	14 (3.9)
Diabetes mellitus, *N* (%)	15 (7.5)	9 (4.2)	17 (4.8)
Pharmacologic treatment	7 (3.5)	8 (3.7)	6 (1.7)
Diabetes and hypertension, *N* (%)	10 (5.0)	5 (2.3)	12 (3.4)

Abbreviations: BMI, body mass index; del, deletion; FISH, fluorescence in-situ hybridisation; GFR, glomerular filtration rate; Ig, immunoglobuline; ISS, International Staging System; RAD, lenalidomide, adriamycin, dexamethasone; t, translocation; VrRd, bortezomib, lenalidomide, and dexamethasone; VCD, bortezomib, cyclophosphamide, and dexamethasone.

**Table 2 cancers-13-01322-t002:** Renal recovery in patients undergoing induction therapy. Estimated glomerular filtration rate at screening in categories. Population: per-protocol set, *N* = 772.

	RAD		VRd		VCD		Overall		*p*-Value (1)
	*N*	%	*N*	%	*N*	%	*N*	%	
eGFR ≤ 50 mL/min	22	10.9	27	12.6	25	7.0	74	9.6	<0.0001
eGFR > 50 mL/min and ≤ 70 mL/min	51	25.2	64	29.9	39	11.0	154	19.9	
eGFR > 70 mL/min	129	63.9	120	56.1	284	79.8	533	69.0	
Unknown	-	-	3	1.4	8	2.2	11	1.4	
Total	202	100	214	100	356	100	772	100	

(1) Chi-square test (2-sided, alpha = 0.05). Abbreviations: eGFR, estimated glomerular filtration rate; RAD, lenalidomide, adriamycin, and dexamethasone; VRd, bortezomib, lenalidomide, and dexamethasone; and VCD, bortezomib, cyclophosphamide, and dexamethasone.

**Table 3 cancers-13-01322-t003:** Renal recovery—estimated glomerular filtration rate (according to Dimopoulos et al., 2010). Population: per-protocol set, patients with eGFR < 50 mL/min at baseline, *N* = 71.

Treatment Arm	*N* in Arm	Renal Recovery Achieved, *N*	%	*p*-Value (1)
RAD	19	12	63.2	0.4747
VRd	27	14	51.9	
VCD	25	17	68.0	

(1) Abbreviations: RAD, lenalidomide, adriamycin, and dexamethasone; VRd, bortezomib, lenalidomide, and dexamethasone; and VCD, bortezomib, cyclophosphamide, and dexamethasone.

**Table 4 cancers-13-01322-t004:** Estimated glomerular filtration rate at screening in the categories (KDIGO 2012 criteria). Population: per-protocol set, *N* = 772.

	RAD		VRd		VCD		Overall		*p*-Value (1)
	*N*	%	*N*	%	*N*	%	*N*	%	
<45 mL/min	17	8.4	21	9.8	18	5.1	56	7.3	0.0428
eGFR ≥ 45 mL/min and <60 mL/min	14	6.9	22	10.3	20	5.6	56	7.3	
eGFR ≥ 60 mL/min	171	84.7	168	78.5	310	87.1	649	84.1	
Unknown	-	-	3	1.4	8	2.2	11	1.4	
Total	202	100	214	100	356	100	772	100	

(1) Abbreviations: eGFR, estimated glomerular filtration rate; RAD, lenalidomide, adriamycin, and dexamethasone; VRd, bortezomib, lenalidomide, and dexamethasone; and VCD, bortezomib, cyclophosphamide, and dexamethasone. KDIGO, Kidney Disease: Improving Global Outcomes

**Table 5 cancers-13-01322-t005:** Renal recovery—estimated glomerular filtration rate after induction therapy in the categories (used by KDIGO 2012). Population: per-protocol set, *N* = 772.

	RAD		VRd		VCD		Overall		*p*-Value (1)
	*N*	%	*N*	%	*N*	%	*N*	%	
<45 mL/min	4	2.0	7	3.3	4	1.1	15	1.9	0.0033
eGFR ≥ 45 mL/min and <60 mL/min	8	4.0	14	6.5	5	1.4	27	3.5	
eGFR ≥ 60 mL/min	182	90.1	176	82.2	341	95.8	699	90.5	
Unknown	8	4.0	17	7.9	6	1.7	31	4.0	
Total	202	100	214	100	356	100	772	100	

(1) Abbreviations: eGFR, estimated glomerular filtration rate; RAD, lenalidomide, adriamycin, and dexamethasone; VRd, bortezomib, lenalidomide, and dexamethasone; and VCD, bortezomib, cyclophosphamide, and dexamethasone. KDIGO, Kidney Disease: Improving Global Outcomes.

## Data Availability

The data presented in this study are available on request from the corresponding author. The data are not publicly available due to the patient-level features involved.

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
