# Peer review of "Kinetics of Renal Function during Induction in Newly Diagnosed Multiple Myeloma: Results of Two Prospective Studies by the German Myeloma Study Group DSMM"

_cancers, 2021, doi:10.3390/cancers13061322_

Round 1

Reviewer 1 Report

Renal recovery and toxicity in triplet therapies used for induction in patients with newly diagnosed multiple myeloma: Results of two prospective studies by the German Myeloma Study Group DSMM

Friederike Bachmann, Martin Schreder, Monika Engelhardt, Christian Langer, Denise Wolleschak, Lars Olof  Mügge, Heinz Dürk, Kerstin Schäfer-Eckart, Igor Wolfgang Blau, Martin Gramatzki, Peter Liebisch, Matthias Grube, Ivana v Metzler, Florian Bassermann, Bernd Metzner, Christoph Röllig, Bernd Hertenstein, Cyrus Khandanpour, Tobias Dechow, Holger Hebart, Wolfram Jung, Sebastian Theurich, Georg Maschmeyer, Hans Salwender, Georg Hess, Max Bittrich, Leo Rasche, Annamaria Brioli, Kai-Uwe Eckardt, Christian Straka, Swantje Held, Hermann Einsele and Stefan Knop

The authors compared renal recovery and toxicity with 3 different triplets used for induction therapy in new diagnosed patients.

Comments

The authors addressed an interesting research question and the manuscript is very well written.

This is a retrospective comparison; hence, it is important to acknowledge the differences in baseline characteristics between different groups, which favored VCD in terms of median GFR at baseline, fewer patients with high-risk cytogenetics and fewer patients with positive urine immune fixation findings. In addition, there were close to two times as many patients with eGFR >30 and <45ml/min in the VRd group (9.8%) as compared to the VCD group (5.1%). Vice versa the proportion of patients with eGFR above 70ml/min was 56.1% in the VRd and 79.9% in the VCD group. The heterogeneity in patient characteristics precludes any meaningful ranking of one treatment over another as they authors did in the last sentence of the abstract and in the sentence before the last in their conclusions.

This limitation has to be considered also in the discussion when the authors address the proportion of patients which improved in the baseline category >30 and >45ml/min. and state that only 1.1% of VCD, 2% of RAD and 3.3% of VRd treated patients remained in this category.

When analyzing the downgrade of renal category, the authors should indicate whether those few patients who experienced downgrading had progression of disease or confounding factors for renal impairment, such as infections, nephrotoxic drugs, bisphosphonates, in order to enable a better understanding of the reasons for downgrading.

Figure 1 and Figure 1a: The number of patients indicated by an individual dot in the figures do not seem to match with the numerical number of patients. In other words there are fewer dots than should be by the number of patients 202 and 214, respectively.

Table 2 has to be interpreted in light of the fact that almost twice as many in the VRD group (9.8%) had eGFR <45ml/min compared the VCD (5.1%) group. The same applies to table 2d when the authors show improvement in eGFR.

The fact that renal recovery was not correlated with depth of myeloma response indicates that renal impairment in many patients was not due to sequels of the disease itself, but rather of the compromised general status of the patient and the fact that they did not enroll patients with stage 4 or 5 of renal failure. For patients of the latter disease category, myeloma response is critical for renal response.

The authors imply that the report by Kumar et al. (Reference 32) is based on prospective randomized data, which is not the case. Hence, they should add that this is a comparison of data obtained by simple chart review, which in addition is limited by short follow-up (less than 2 years).

Some of the tables could be omitted.

Author Response

Pls. refer to the attached document.

Reviewer 2 Report

In this study, the authors compared the renal response of newly diagnosed myeloma patients treated with VCD from a phase 2 study and a phase 3 study comparing VRd vs RAD. The comparison was made using the IMWG renal response criteria, eGFR slope and improvement in CKD stage migration. The authors found that treatment with VCD was associated with the highest rate of renal CR while VRd had the lowest. However, positive CKD stage migration was seen most with VRd and least with VCD. This was also true with negative CKD migration where VRd had the highest and VCD had the lowest. Slopes of eGFR were highest for patients treated with VCD and lowest with VRd. Interestingly, the authors did not find an association between hematologic response and renal response.

While the paper was well written and much work went into it, the applicability and the analyses performed are a bit difficult to interpret. First, due to the eligibility criteria of the studies, majority of the patients had near normal kidney function. Only 9.6% of the patients had an eGFR of < 50 ml/min/cor and 7.3% were < 45 ml/min/cor. This is much smaller than what is reported in the general MM population which is typically between 20-40%. In addition, these are not the patients that have AKI because the lowest eGFR in the study was only 24. These are the patients that are more likely to have CKD or acute kidney disease where the decline in eGFR is over months rather than days to weeks. These patients may have AL amyloidosis, monoclonal immunoglobulin deposition disease, light chain proximal tubulopathy, etc. The renal recovery criteria from the IMWG was really made for patients with acute kidney injury most likely secondary to cast nephropathy.   Application in the current population is not validated and the term renal recovery here is misleading.

Second, using renal recovery (CR) by IMWG criteria where the eGFR needs to go from < 50 to > 60 is difficult to interpret when the average eGFR for the groups is 73.1 – 95.6. Improvement by CKD stage migration is even harder to interpret since a patient can have a 14 ml/min improvement from 30 ml/min and not have a stage migration while another patient could have a 5 ml/min and have a stage improvement. Not only that, a 5 ml/min increase from 56 ml/min is a 9% improvement in eGFR but is a stage improvement while a 14 ml/min increase from 30 is a 50% improvement but no stage improvement. A better assessment would be straight % improvement or decline in eGFR. This might help explain why VRd was associated with the most improvement and decline in stage migration. It would also help to know what was the range of the change in eGFR.

The biggest question with this study is how do we interpret the results? Kidney function is important in MM because it is a prognostic factor. Did the differences in renal function improvement affect outcomes of these patients? The authors hinted at the fact that VCD may be the better choice in patients with renal impairment but this is only based on a higher percentage of renal CR which we do not know what it means? Did the patients with the renal CR have better or worse outcomes as compared to patients who did not achieve renal CR or patients who did not have renal impairment? It is always difficult to compare outcomes across studies but there should be enough patients to the comparison within each study.

The authors compared renal response with hematologic response but they picked > or < PR as the cutoff. Generally, VGPR is the hematologic response that seems to be associated with renal outcomes. I would suggest repeating the analysis with VGPR.

Author Response

Pls. refer to the attached document.

Reviewer 3 Report

Review of the manuscript entitled "Renal recovery and toxicity in triplet therapies used for induction in patients with newly diagnosed multiple myeloma: Results of two prospective by the German Myeloma StudyGroup DSMM” authored by Friederike Bachmann et al.

Authors aim to compare kidney-specific variables (pre- and postinduction renal function) in a large cohort of myeloma patients - two prospective German studies (DSMM XI and XIV)- receiving three different three-drug regimens for initial myeloma treatment. All regimens had a positive impact on kidney function without a difference in the proportion of patients who reached normal renal function after each three cycles. Interestingly, patients who received bortezomib, lenalidomide and dexamethasone had a higher risk for a worse renal function following induction, when compared to initial values.

The study design and description of the variables of the study are well indicated. Monitorization of the renal parameters is well described and results well summarized.

Statistical analysis seems to be adequate for the analysis of variables.

The baseline characteristics of the participants is well documented, together with relevant comorbidities.

Authors also indicate limitations of the study.

The study is well conducted and easy to follow. The manuscript is well written and organized.

The conclusions are supported by the analysis performed.

In general, I find the findings very interesting and that could be relevant and valuable in a clinical setting.

Minor comment:

- In the discussion, it should be indicated the added value of the results of the study compared to other similar studies if any.

Author Response

Pls. refer to the attached document.

Round 2

Reviewer 1 Report

Basically I recommended to accept the manuscript in its present form